# Situated AR Simulations of a Lantern Festival Using a Smartphone and LiDAR-Based 3D Models

**Naai-Jung Shih** [1,*] , **Pei-Huang Diao** [2] , **Yi-Ting Qiu** [1] **and Tzu-Yu Chen** [1]

1   Department of Architecture, National Taiwan University of Science and Technology, 43, Taipei 106, Taiwan; chooi0215@gmail.com (Y.-T.Q.); ziyouchen.0617@gmail.com (T.-Y.C.)
2   Department of Advertising, School of Art and Design, Guangdong University of Finance and Economics, Guangzhou 510320, China; phd@gdufe.edu.cn
*   Correspondence: shihnj@mail.ntust.edu.tw; Tel.: +88-6022-7376-718

**Abstract:** A lantern festival was 3D-scanned to elucidate its unique complexity and cultural identity in terms of Intangible Cultural Heritage (ICH). Three augmented reality (AR) instancing scenarios were applied to the converted scanned data from an interaction to the entire site; a forward additive instancing and interactions with a pre-defined model layout. The novelty and contributions of this study are three-fold: documentation, development of an AR app for situated tasks, and AR verification. We presented ready-made and customized smartphone apps for AR verification to extend the model's elaboration of different site contexts. Both were applied to assess their feasibility in the restructuring and management of the scene. The apps were implemented under a homogeneous and heterogeneous combination of contexts, originating from an as-built event description to a remote site as a sustainable cultural effort. A second reconstruction of screenshots in an AR loop process of interaction, reconstruction, and confirmation verification was also made to study the manipulated result in 3D prints.

**Keywords:** AR; LiDAR; RP; smartphone; 3D sustainability; lantern festival; intangible cultural heritage





## 1. Introduction

Festivals have been categorized as a form of Intangible Cultural Heritage (ICH) [1–3] and promoted by different governments, organizations, and media [4–7]. A lantern festival not only constitutes an important cultural asset, but also creates dynamic interactions with the urban fabric. The festival setting generates a temporary fabric, which is usually removed after the event and deserves both documentation and dissemination. For physical installations, related studies have been undertaken focusing on photogrammetry modeling [8], interactive systems [9], augmented reality (AR)/virtual reality (VR)/mixed reality (MR) [10], and 3D printing [11], in which devices, such as 3D scanners [12–14], low-cost 3D scans using Google Tango® (2014, Google Inc., Mountain View, CA, USA) tablets [15,16], smartphones or handheld devices [17], and unmanned aerial vehicles (UAV) and light detection and ranging (LiDAR) [18], have been applied with different levels of efficiency and effectiveness. The photogrammetry models were created by applying multi-sourcing [19] or crowdsourcing [20], or in a collaborative manner [21]. Moreover, virtual humans [22], virtual crowds [23], large populations [24], video data of real actors [25], and cartoon-like characters [26] were applied to attract users' interest or as a way to promote communication. These applications have been proven to be feasible for varied cases [27–31] and scales [32,33], with awareness of location and context [34]. The documentation is related to information systems [35] and as part of a communication model [36] for exploration and dissemination [37]. Engagement and learning were also assessed [38].

Point clouds have been applied to create mesh models for AR application [39]. Current technology already enables automatic documentation and registration at the size of a backpack for terrains that are unreachable by vehicle. Although Light Detection and

Ranging (LiDAR) scans can apply a Global Positioning System (GPS) and Inertial Measurement Unit (IMU) on a vehicle for city model reconstruction [40], a static scan is usually applied for a superior scan and registration result. The modeling effort in conversion, meshing, and hole-filling should be alleviated by the direct utilization of a point cloud in AR, regardless of its limits in transparency, larger file size, and the interference of other objects during the scan process. Indeed, a well-sampled and well-textured point cloud model provides a convincing effect with the high resolution of an as-built scene.

AR-related applications have been successfully implemented in navigation, education, industry, and medical practice [41–44]. They combine, register, and interact with real and virtual information in a real environment [45–47]. Reality with an interactive and highly dynamic experience can be explored using mobile AR applications for layers of information [48,49]. Considering a case study of a local lantern festival formerly applied to a communication model [50], engagement can be facilitated with improved comprehension through interactions with 3D scenes or entities in AR. To recall a festival that comprises a complicated setting, AR should be applied so that the representation of the past virtual environment can be enriched to achieve a vivid tourism experience.

AR applies 3D models in different environments for meaningful context elaboration. The interactive models are usually applied to different sites to fulfill aims in pedagogy or professional practice. The models, which are presented as part of a general structure with no specific preference or constraint, may require re-construction or re-framing of a scene for a specific task. Management issues arise for models, or their instances, with matching backgrounds. The reconstruction of appropriate scenes has proven to be highly beneficial in pedagogy with real-time streaming of contents. Mixed Reality (VR and AR) has also been applied for cultural heritage for the benefits of both [51]. Although post-processing needs to be applied for a higher reality effect, direct collaboration in a real environment using AR can be more straightforward in connecting the temporal experience of an event between different locations and periods of time.

The proposed 3D data acquisition was conducted in the evening. The 3D laser scan constitutes an effective tool for acquiring geometric attributes, such as detailed trees and vegetation [52]. For night-time road and street environments, lighting conditions through 3D point clouds were evaluated [53]. A previous investigation used a 3D scan for a special urban evening event in a city and a festival [54]. Although laser scanners in different illumination conditions were assessed for accuracy [55], structural details still need to be verified in low luminance conditions. However, 3D scanning laser-rangefinders for visual navigation techniques could be made by applying raw laser intensity data into greyscale camera-like images [56].

## 2. Research Purpose and Methodology

For a lantern festival held in the evening with crowds, the current study aimed to determine if large-scale documented 3D data can be applied in AR interactions or to assist planning [57] in professional design practice. Moreover, we aim to investigated whether various planning flexibilities of AR interactions can be conducted in a ready-made or customized environment, whether convenient mobile devices, such as smartphones, can be applied in AR, and whether human characters can be applied as a part of a scene by increasing the involvement of the surrounding crowds in AR.

This research intended to represent and reconstruct a cultural festival. The representation required appropriate 3D documentation to retrieve and identify significant settings of the scene. The reconstruction of old scenes concerns related context applications in terms of the correlations and interactions of installations in different types and scales. The exemplification should also be applicable to traditional festivals held in the past.

We proposed LiDAR-based 3D data and an AR app for supporting documentation and interaction in the representation and reconstruction. The 3D entities should be interacted with using a convenient smartphone and app for different scenarios with matching contexts. Cloud access of a ready-made decimated point cloud or mesh models should enable the



app to be implemented under a homogeneous and heterogeneous combination of contexts, as an extension to 3D sustainability.

This study should not only develop an app, but also restructure a cultural event for the feasibility of context adaption. The representation of cultural heritage as a single object differs from that of an event, which consists of urban fabrics, installations, facilities, supporting structures, and visitors. The introduction of an entity should be represented with a collection of multiple correlations in an arrangement of a spatial structure. The correlation forms the complexity of the scene and the AR to be applied.

Challenges exist in presenting a contemporaneous traditional event, concerning how to define related elements, establish correlations, and create media to determine pertinent relationships between factors. As-built data from real scenes should be utilized to support the illustration of details and spatial structure to a level that establishes a synergistic relationship with the AR application. Reverse engineering should also be implemented with approaches, processes, and expected outcomes to interpret the unique case of a festival, instead of inspecting original design plans.

A verification of the simulation was performed by the reconstruction of a 3D model in an AR loop process of interaction, reconstruction, and confirmation. A series of screenshots were recorded for the second photogrammetric modeling of a 3D printed model to study the manipulated result. The novelty of this research was the application of AR in a cultural festival, with data originating from a relatively large size of point cloud. This app should offer models with higher documented reality. The 3D entities should be interacted with using a convenient smartphone and app for different scenarios with matching contexts. Tests by the researchers should be conducted at the original site and in a different locations as a situated application. Test opinions should give the verification of the applications, such as re-experiencing a former event, enabling the display or 3D layout of the context like a canopy with an on/off option, and its successes and restrictions.

One of the main purposes of this article is to discuss the feasibility of the application, and the survey of user experience will be the focus of the second stage in the future.

This paper is organized in the following sections: introduction, research purpose and methodology, background of a lantern festival, restructuring and management of AR instancing, implementation of the 3D scan data, three AR instancing scenarios, verification by the situated application and reconstruction of a 3D model, discussions, and conclusions.

### 3. Background of a Lantern Festival

A lantern festival was selected for its unique complexity, cultural identity, and role in memory. The festival is an important Chinese event held immediately after the Lunar New Year. It features enormous collections of installations inspired by folk stories and/or traditional architecture. The installations create a remarkable night scene and attract visitors island-wide (Figure 1a). The event has a major theme lantern of the year with designated subjects distributed in various zones. Large-scaled displays have become a regional highlight and have promote tourism. This extraordinary urban evening event represents a planning of a display and interaction with traditional icons and customs.

The festival only lasts for approximately two weeks. A feasible application of model instancing in different urban contexts should be provided to fulfill a recall or promotion attempt. For a short-term festival event with a large number of installations, a 3D scan provides a rapid and detailed architectural documentation of the lanterns and the spatial structures (Figure 1b,c). A system should be developed to integrate the documented data and a convenient device to represent the festival with a co-existing relationship of entities. The 3D data are important in the construction of spatial frameworks, so that a tourism-like experience can be constructed from linear axes with relative reference made between the main lantern, smaller installations, visitors, and supporting constructions.

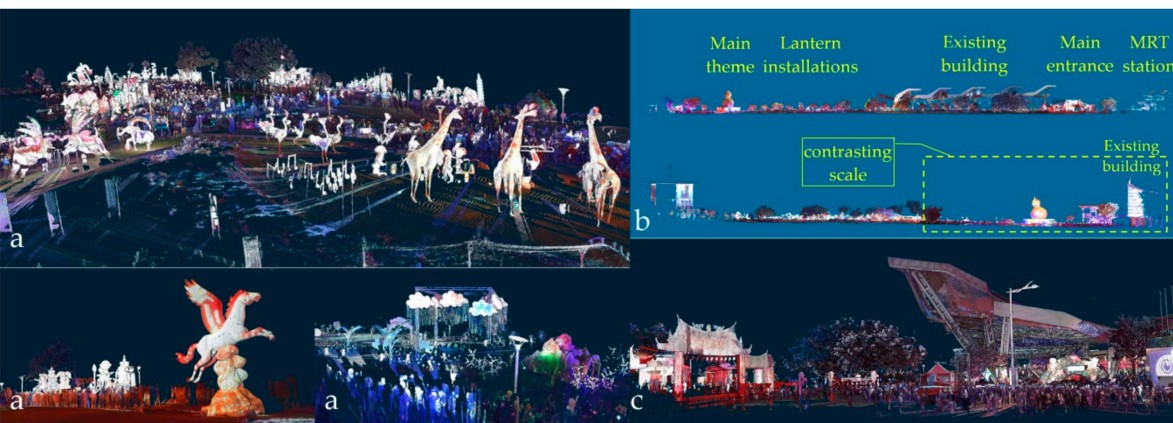

**Figure 1.** (**a**) The 2016 Taipei Lantern Festival with different types of installations; (**b**) the spatial structure; (**c**) the main entrance to the main theme lantern using the point cloud model.

## 4. Restructuring and Management of AR Instancing

An AR app works differently with the management of instances and backgrounds before and after instancing. An interaction starts by selecting models and manipulating backgrounds for subjective study afterwards. Then, a model has to work with its relationship to other models. Assistance may be required to anchor or align one to another. Management is different from instancing in a general AR application with a free selection of elements for a room or a space, since a problem space is defined by specific elements (e.g., a furniture set) and a constrained operation of relation location.

A bottom-up or top-down metaphor is presented in the restructuring and management process. A scene is usually created as a bottom-up process from a sequential instancing of the individual model to the final scene. All of the models come from categories that concern the same hierarchy. A general purposed off-the-shelf app or predefined platform, such as Sketchfab® (2020, Sketchfab, New York, NY, USA) [58] or Augment® (2020, Augment, Paris, France) [59], usually supports multiple categories of 3D models. The instances present a general structure with no specific preferences or constraints in application.

The structuring and management of AR models are different using the 3D scan process. For a festival with 3D scans, the individual scans are registered into a whole as a bottom-up process. Each model object must be separated from the registered point cloud model in a top-down differentiation process. Both processes construct and restructure a hierarchy. Irrespective of how a scene is organized, a group of instances is initially subject to a certain organization of framework. The restructuring and re-framing of a scene involves, for example, the grouping or ungrouping of specific instances of a task. Moreover, each instance may have dissimilar degrees of freedom. For instance, an interior design may have applied 4-degrees of freedom (3-way transition + Z-axis rotation) to each piece of furniture, fix-shaped installation, or facility.

AR requires the restructuring and re-framing of a scene with an appropriate switch at the macro- or micro-scope. Comparing the constrained and free manipulation of instances, a scene should be initiated with a top-down decomposition of elements for bottom-up context matching. The switch between a macro- and micro-perspective is crucial, not just in scaling the scene for appropriate visibility, but also in the restructuring of a scene and elements to support design comprehension and development in the different backgrounds of urban fabrics or objects with cultural identities. In real applications, the macro-view presents the application, Sketchfab®, to the full scope of the point cloud. The micro-view, on the other hand, presents the development of an app using ARKit® (v1.5, Apple, Cupertino, CA, USA) [60] or the application, Augment®. The former is a customized, domain-specific, and design-specific application with a constrained arrangement of instances; the latter constitutes a general-purpose app with general support of instancing arrangements.

## 5. Implementation of the 3D Scan Data

The 3D data were implemented for cultural preservation, context extension, and the spatial structure analysis of a sustainable festival. The preservation was conducted with more of a direct connection to our daily life, in which the memory should be reserved and recalled. The presentation of structure was exemplified by the definition of spatial frames and elements, such as the canopy and ground level, and infills, including installations and crowds. The structure led to an appropriate representational hierarchy of the scenes to be displayed. Indeed, any part of the framework or lantern installation was feasible for a homogeneous and heterogeneous combination of contexts.

The mesh model of multiple objects or the point cloud of the entire site practically correlate the data in different hierarchies from field work and laboratory experiments to different sites. Both data types were able to be interacted with. The app application and development followed the layout of objects and the spatial structure of an event. Since the off-site instancing experience is to be extended to other architectural spaces or urban fabrics, explorations were made of different development approaches, data retrieval, format conversions, app platforms, and instancing scenarios.

3D scan data were utilized to define festival entities and their correlations, both visually and physically. Faro Focus 3D® (FARO Technologies Inc., Stuttgart, Germany) was used to scan the 2016 Taipei Lantern Festival. Field work included 37 scan locations, in which more than 19 GB of data or 543 million points were retrieved in a region approximately 300 m wide and 250 m long. The subsequent data manipulation effort constituted more than 10 times that of the field effort. A detailed description and figures of the methods applied can be seen in Figure 2, which included registration inspection, filtering and decimation, scene segmentation, texture adjustment, and format conversion. It is worth noting that holes were purposely not filled to prevent misleading outcomes of the configuration in over-crowded scans.

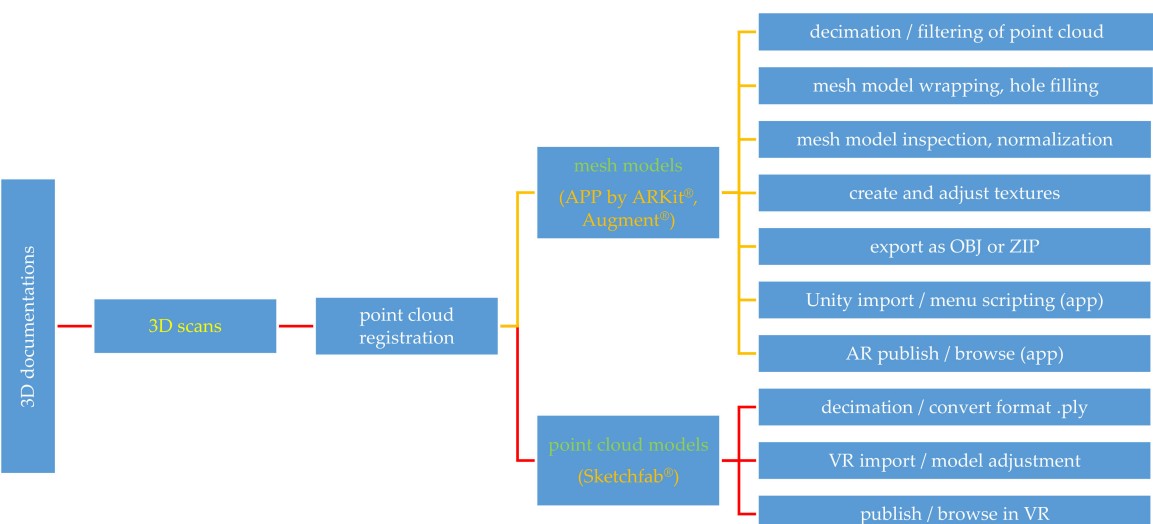

**Figure 2.** The data flowchart.

Two approaches were utilized. The first one used a public platform to engage the point cloud as the type of data to interactively manipulate in VR. The second was an app developed specifically for the mesh model, converted from the point cloud in AR. Both types of interactions have to be made on a smartphone platform for the different scenarios of site contexts.

The point cloud model was also able to create a mesh model for AR user interaction (Figure 3). Although photogrammetry can create a large area of terrain model using images taken by an Unmanned Aerial Vehicle (UAV) or Unmanned Aircraft Systems

(UAS), the mid-range 3D scanner captures small objects, such as power wires, stage frames, lamp poles, tree branches (Figure 4), and installations on building façades with sufficient detail to facilitate inspection. In addition, crowds were also used as a specific element for the reference of scale (Figure 5), and the real depth was usually applied by Depth API (ARCore 1.18, Google Inc., Mountain View, CA, USA) [61].

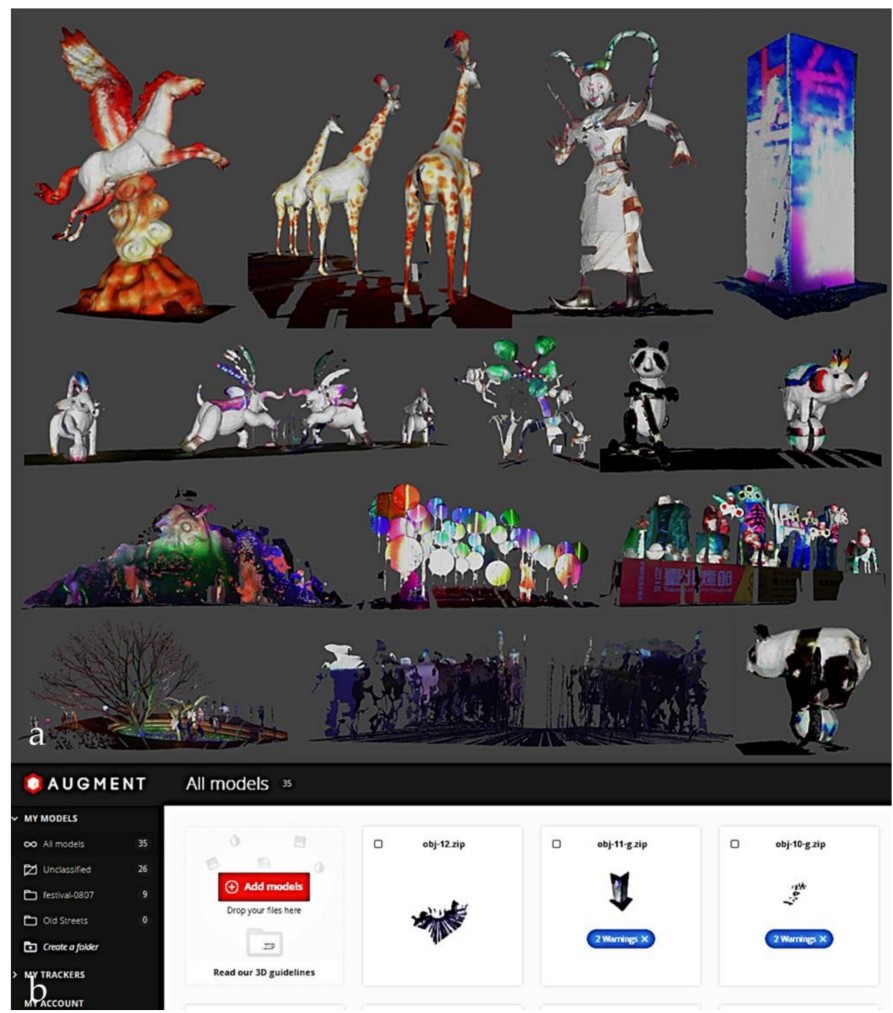

**Figure 3.** (**a**) Augmented reality (AR) models for situated simulations; (**b**) corresponding representations in the augmented reality (AR) app.

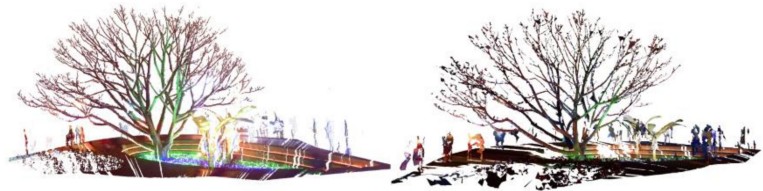

**Figure 4.** Original scan: 1.38 million points vs. Augment® AR mesh: 0.28 million points, 0.50 million polygons.

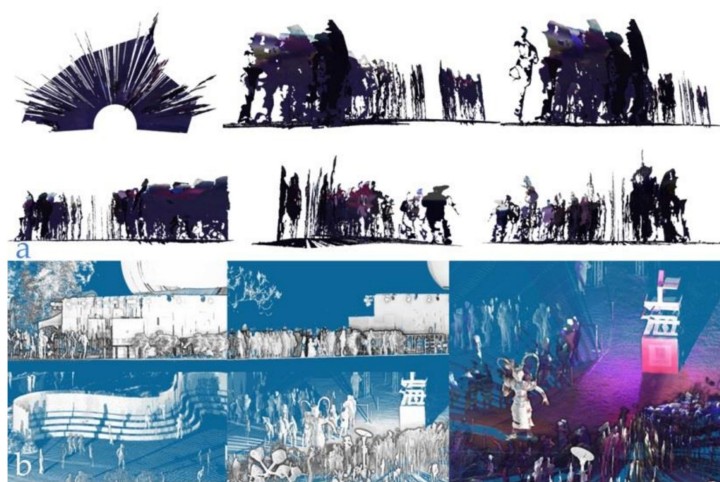

**Figure 5.** (**a**) The crowd appeared as multiple moving trails in radical shape around the scanner; (**b**) grouping configurations.

The model worked with software, including CloudCompare® (v2.6.1, EDF R&D, Paris, French) [62], Geomagic Studio® (v2014.1.0, former Raindrop Geomagic Inc., Morrisville, NC, USA, now 3D Systems Company, Rock Hill, SC, USA) [63], Meshlab® (v1.3.3, Visual Computing Lab–ISTI–CNR, Pisa, Italy) [64], and Autodesk Revit® (2020, Autodesk Inc., San Rafael, CA, USA) [65], to review the model, convert formats, and retrieve sections, skylines, ground plans, and images. In addition to the color illustrated on 3D point clouds, an Eye Dome Lighting (EDL) rendering process [66] was applied to differentiate physical elements, particularly in a colorless mode for a better display of edges, depths, or silhouettes (Figure 5b) during the study process.

Among two major AR types [67], markerless AR contributes to more diversified application scenarios [68,69], concerning the difficulty involved in arranging markers pasted on building surfaces in marker-based AR. A markerless smartphone AR app was developed for convenient interactions with lanterns and facilities.

## 6. Three AR Instancing Scenarios

Three AR instancing scenarios were applied (Figure 6) based on whether a scene could be manipulated at the element level and how the manipulation was conducted (with or without a pre-defined model structure in the initial state of interaction). The tests were conducted only from the interaction made to an entire scene using an existing app, and to an individual model within a pre-defined layout using a customized app. A homogeneous combination of backgrounds refers to a plain one, or the same site as a festival recall. A heterogeneous combination refers to different backgrounds in the context or site of study, promotion, or design development.

The three scenarios were applied. The first two were conducted as tests, and led to the development of the last one as a customized app to represent a previous lantern festival. The tests were conducted in four places: a laboratory, the original festival site, a remote site, and a personal working space.

- The first: to browse. An interaction with the entire site was made. The point cloud of the entire festival scene was decimated, viewed, and interacted with in VR, but no sufficient support of point data conversion to AR was provided.
- The second: to interact and interpret. A forward additive instancing and interaction was made. A general set of instances was used for context-matching from individual elements to a scene. Mesh models were created and were effective in free-instancing. However, no sufficient support to the on/off display option or assistance to precisely controlled the relative location were provided. Different types of background context were applied as needed.

- The third: to recall, interact, interpret, switch display, and evaluate. An interaction with the pre-defined model layout was made. A structure with a set of pre-loaded instances was used for the recall of former lantern settings. The customized app had the relative location of models pre-defined as a reference of familiar scenes. Context matching was carried out from a remote site. Design alternatives were supported with on/off options for each model for evaluation. Different types of background context were applied as needed.

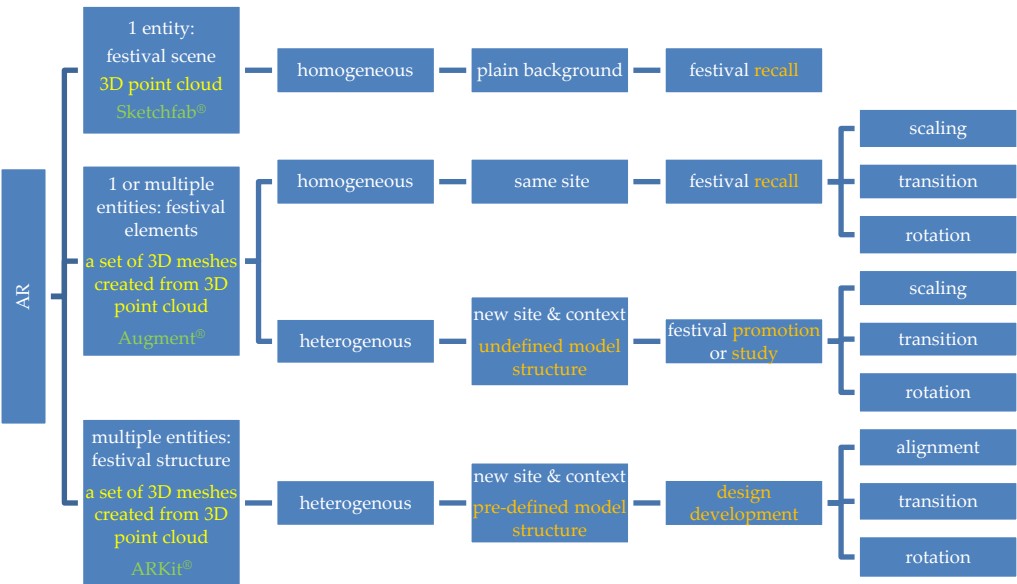

**Figure 6.** Three scenarios of AR instancing and corresponding data and manipulation.

### 6.1. An Interaction with the Entire Site

For the comparison of the application environment, a smartphone with Android 8.1 (Google Inc., Mountain View, CA, USA) or later was adequate to display individual entities using Sketchfab® without menu customization. It allowed multiple cloud downloads of 3D models in mesh or point cloud format. Instead of restructuring the scene, the entire scan was used (Figure 7a–c). The original registered file was 19 GB in PLY format. It was decimated to 190 MB in 12.2 million points (truncated from 22 million by the app), which is just under the 200 MB limit of the subscribed plan. The 3D point cloud model in AR was not supported. For a smaller size of mesh model, the mesh model in Augment® (Figure 7d) was also able to be interacted with (Figure 7e).

Trade-offs existed between the details, file size, and the scope of the scene. The size was approximately two-thirds of the app developed by ARKit® of the third scenario, irrespective of whether or not the entire scene was available. A more direct application of the point cloud was made possible without segmentation into parts. A more connected working environment was achieved between the manipulations prior to the AR conversion and afterwards. The full resolution of the point cloud before decimation can identify the scale of crowds from small particles to curved silhouettes (Figure 7c,f).

One of the axes in Figure 1b was applied to illustrate the main theme lantern and a group of purposely modeled visitor configurations for context adaption. Users were encouraged to relocate themselves and run the app in different contexts to make comparisons.

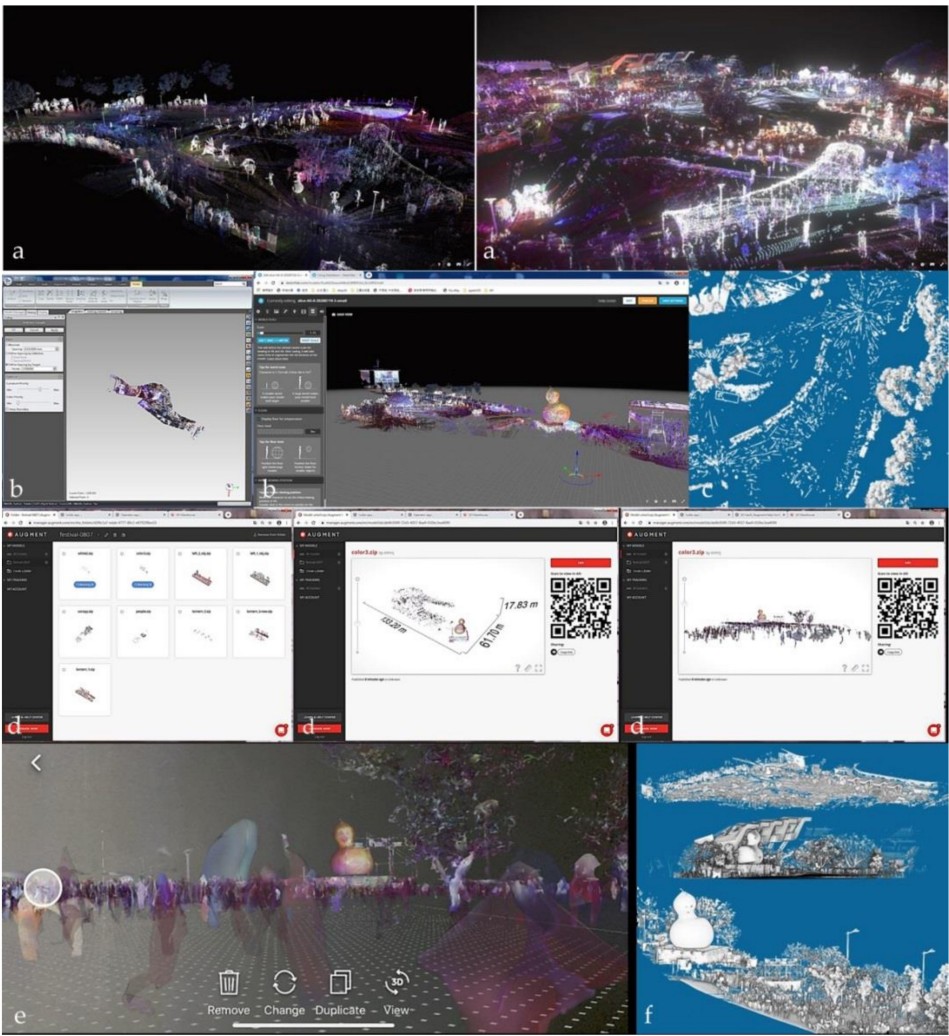

**Figure 7.** (**a**) Screen shots of the Sketchfab® app in browse and virtual reality (VR) mode; (**b**) converting process for decimation and model setting; (**c**) a full resolution with the silhouettes of crowds identifiable when a small part was presented; (**d**) the Augment models; (**e**) the decimated mesh model in Augment®; (**f**) the point cloud model in Meshlab®.

### 6.2. A Forward Additive Instancing and Interaction

The smartphone AR app, Augment®, was used for individual instancing of the components of the spatial structure presented in the second scenario. Each lantern was manipulated as needed in different urban fabrics or in drawings when a matching context was selected. The models were re-installed back to the original site as a recall of a former event (Figure 8). The interaction was mainly recorded in the evening by screenshots, third-person photographs, and the stepwise operation process (Figure 8d). Each model can be relocated or scaled based on the relative proportion to the researcher for a personal and flexible interpretation of the design. The interaction also brought people's attention to how a former temporary fabric had enriched the activity of a public open space with cultural diversities.

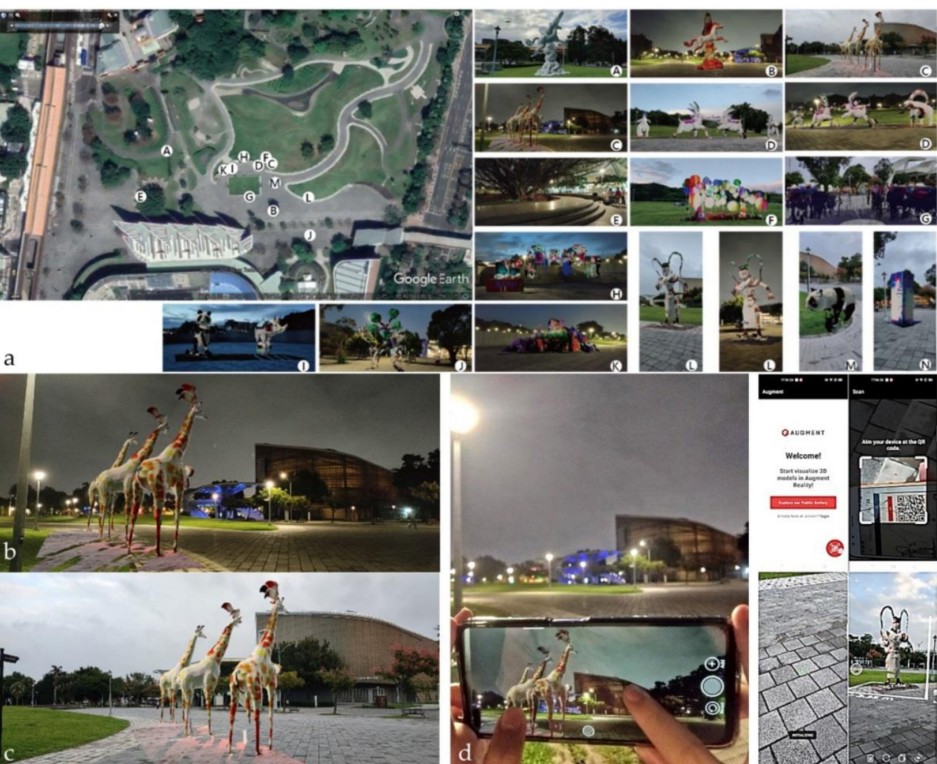

**Figure 8.** The situated revisit and recall of an old festival with AR screenshots (**a**), screenshots in the evening and late afternoon (**b**), third-person photographs (**c**), and the stepwise operation process: launch app, scan QR code of a selected 3D model, detect plane, tap to allocate the model, and adjust the model's scale, location, or rotation angles (**d**).

For a group layout tested in the laboratory, all of the model instances created a working space without a predefined relative location to each other. Although the working space can be shared and recalled afterwards, the initial state was created from scratch as part of a forward design loop. In particular, a model could not be displayed on/off individually as needed for layout evaluation unless it was deleted. This on/off option, which is important in the presentation of design elements, was changed in the customized app in the following section.

### 6.3. An Interaction with a Pre-Defined Model Layout

The meta-relationship between elements was expected, with identified arrangements between people, installations, temporary fabrics, and permanent fabrics. The developing steps and the structural diagram of elements of the AR app can be seen in Figure 9. The markerless AR app was created from Unity® (2020, Unity Technologies, San Francisco, Cupertino, CA, USA) using ARKit SDK® (2020, Apple Inc., Cupertino, CA, USA). Visual Studio® (2020, Microsoft, Redmond, WA, USA) was applied to edit the display and switch the virtual models. The menu to the left of the screen could be retracted as needed. The interacted items and system interface of the app included the selection of a series of objects. All of the contents could be selected individually or accumulatively. It only required a surface, such as the ground level, to anchor the 3D models, instead of a barcode or a picture. The markerless app made the application operate more straightforwardly, without being restricted to additional marker-paste procedures or potentially causing damage to the targeted surface.

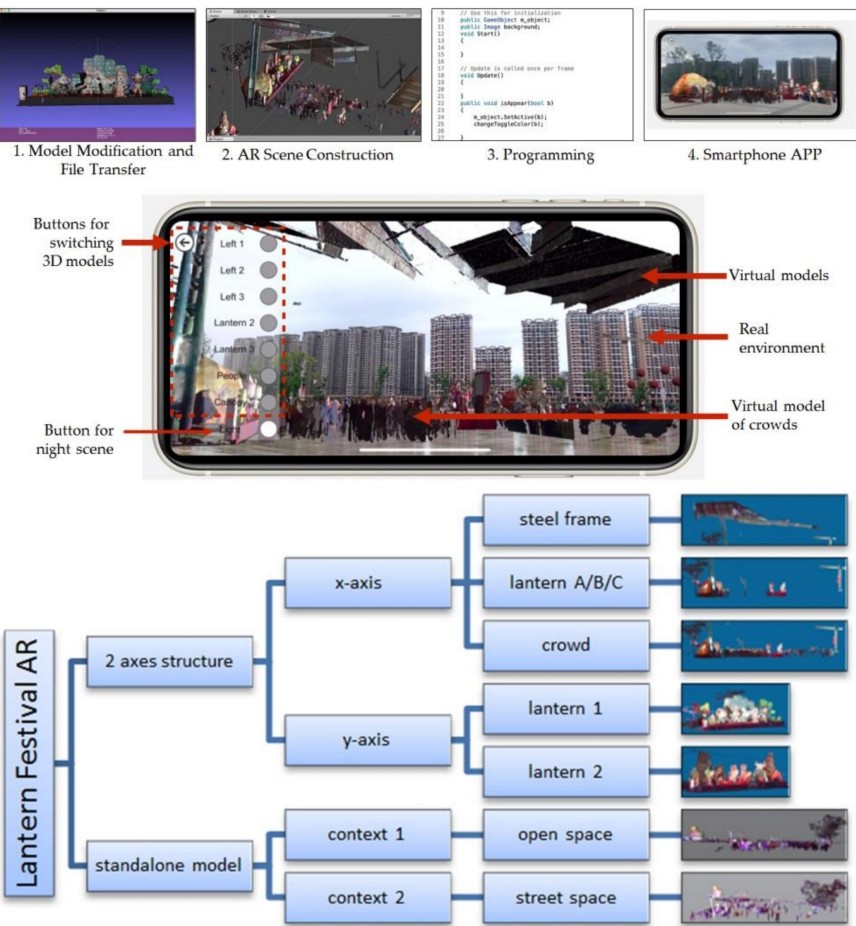

**Figure 9.** The development process of the AR APP (**top**), smartphone interface (**middle**), and the structural entities of the lantern festival (**bottom**).

An iPhone XS Max® (A12 Bionic chip, 64G Capacity, Apple, Cupertino, CA, USA), which was equipped with an A12 CPU with 4 GB RAM and 64 GB internal storage, was used to test the app. The final size of the app was 288.5 MB. Due to the limited storage space of a smartphone, increasing the polygon number caused a delay in the screen display, with flickering. Decimation was made by the defined polygon count or percentage. The cloud model was converted to a mesh model in OBJ format and decimated using Geomagic Studio® (v2014.1.0, former Raindrop Geomagic Inc., Morrisville, NC, USA, now 3D Systems Company, Rock Hill, SC, USA). For display efficiency, the original mesh model of 121 MB (0.73–62.70 MB each) was divided into seven parts along two linear and perpendicular axes (Figure 10a). Each decimated part was approximately 88,106–887,408 polygons. The standalone mesh model was 171.1 MB, comprising approximately half of the polygon numbers wrapped afterwards from the original scanned points. The axes represented one of many possible combinations that visitors perceived in different perspectives (Figure 10).

The site planning, which was subjected to the programming of the festival, created a hierarchical space based on the management of two-axis spatial structures for lanterns and facilities (Figure 11). The structure constituted a semi-open arcade which consisted of the arrangement of a roof, visitors, and three groups of lanterns. One of the three groups of lanterns was selected to form another axis with two additional instances to be interacted with upon selection on the menu. The app display scenario was constructed based on the as-built spatial structure, which is similar to what occurred to the adjacent fabric in an existing 3D spatial framework in reality. Lantern installations and crowds can

be displayed selectively to reveal the scale factor of the surrounding environment, from a smaller installation to a large canopy.

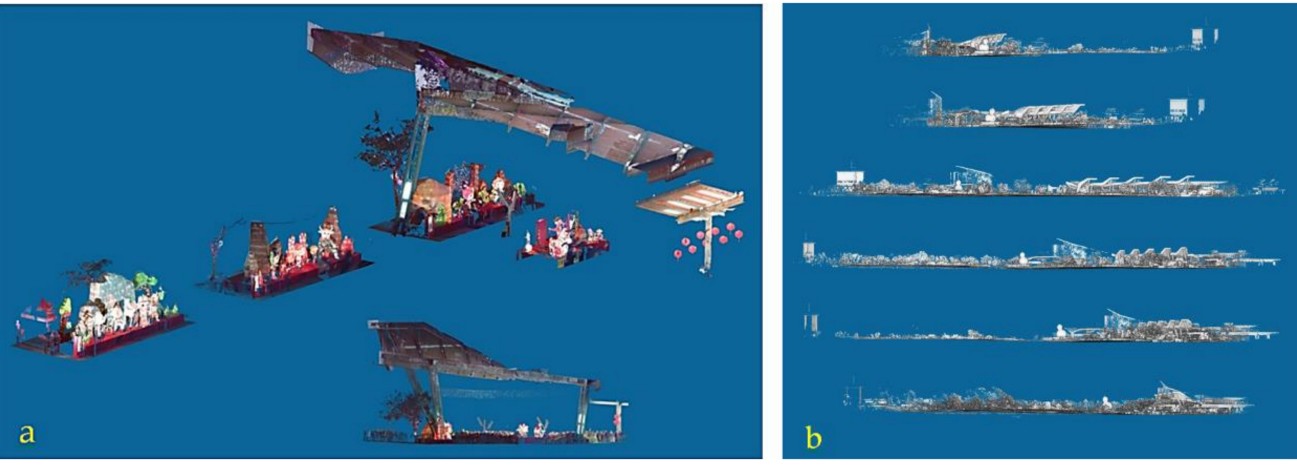

**Figure 10.** Two axes of lantern models (**a**) and many possible combinations perceived by visitors from different orientations (**b**).

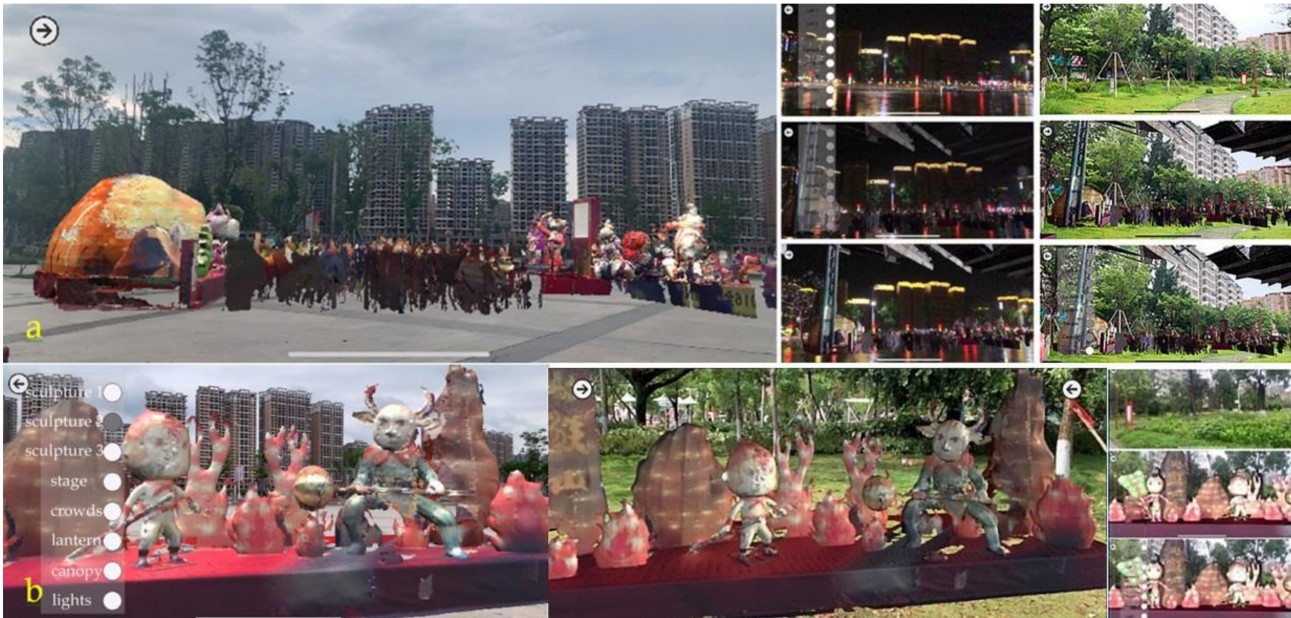

**Figure 11.** The two-axis spatial structure (**a**) or individual lantern (**b**) was combined with different real environmental contexts in the evening or in the daytime.

An event-specific or occasion-related app should be designed to enable 3D scenes with unlimited adaption outside of the original display scenario. Our app was designed with the individual scenes separated, so that it can be carried around to project the standalone 3D model in different urban fabrics as a correspondence to different cultural contexts or as a feasibility test of different environmental settings. Since this app does not require markers, the designated models are suggested to be projected at ground level to merge with existing urban contexts as it shows on screen, such as different landscapes, architecture, streets, or open spaces. Indeed, a user can walk inside a virtual crowd and feel the relative proportion with the theme lantern, just like in reality. The main theme can be relocated and

evaluated in a more diversified arrangement of urban space, or next to the theme lantern next year as a connection to a past memory to promote festival sustainability.

The on/off display is a typical function commonly applied in drafting or modeling environments. Many tools were applied to achieve this through, for example, the option to control layer visibility. However, none of these tools should delete the model, since a model represents a trail of former design and a record for future reference. This is particularly the case when a modification was made inside of an environment replete with as-built facilities. The modification should start with a set of models called from a former version of a design plan, and be followed by an on/off display switch to reveal their correlation to the spatial structure. The display support, the individual manipulation options, and a pre-defined structure of models made the app a diversified design tool for management.

Combining AR and a 3D model of night scans achieved a captivating experience. The interaction with entities created an extended system of point cloud data. Scenes are better shown in the evening or in a room with the lights turned off. The app can also be used in daylight for the design layout evaluation of different sites. The crowds, which are usually avoided for a clear display of a scene, were purposely retained as a scale reference. It allows a user to walk into the crowds, exactly as it was experienced in the festival.

The validation of the implementation was primarily made with field tests of multiple urban contexts or cross-regions from different locations. Other than the weather issue that was present in the outdoor application, the app was proven to be feasible for indoor browsing at any time. Due to the file size being up to 190 MB, a capable bit stream that connects to the cloud should be provided. A 4G network was used and revealed that the benchmark of browsing experience was acceptable.

## 7. Verification by Situated Application and Reconstruction of the 3D Model

The verification was made by a situated application and the reconstruction of a 3D model in a loop process (Figure 12) of interaction, reconstruction, and confirmation. The AR interaction from different orientations was recorded as a series of screenshots for photogrammetric modeling. The reconstruction from interaction created a more solid integration of the context during the situated study. To confirm the manipulated result, the AR simulation was substantiated and further verified by a 3D printed model (Figure 13).

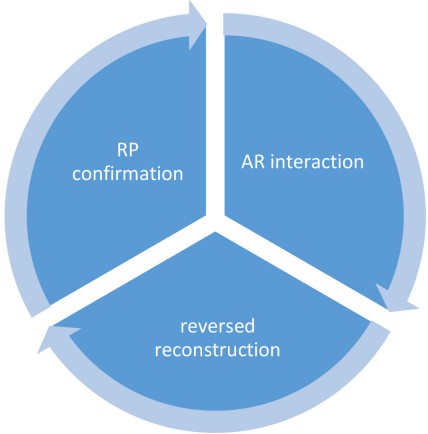

**Figure 12.** The AR interaction, reversed reconstruction, and rapid prototyping (RP) confirmation process.

We found that rapid prototyping (RP) confirmation enabled a broader interpretation of lantern installations with context, based on personal preference. The interpretation brought back the former entity and facilitated a more flexible elaboration of the original design.

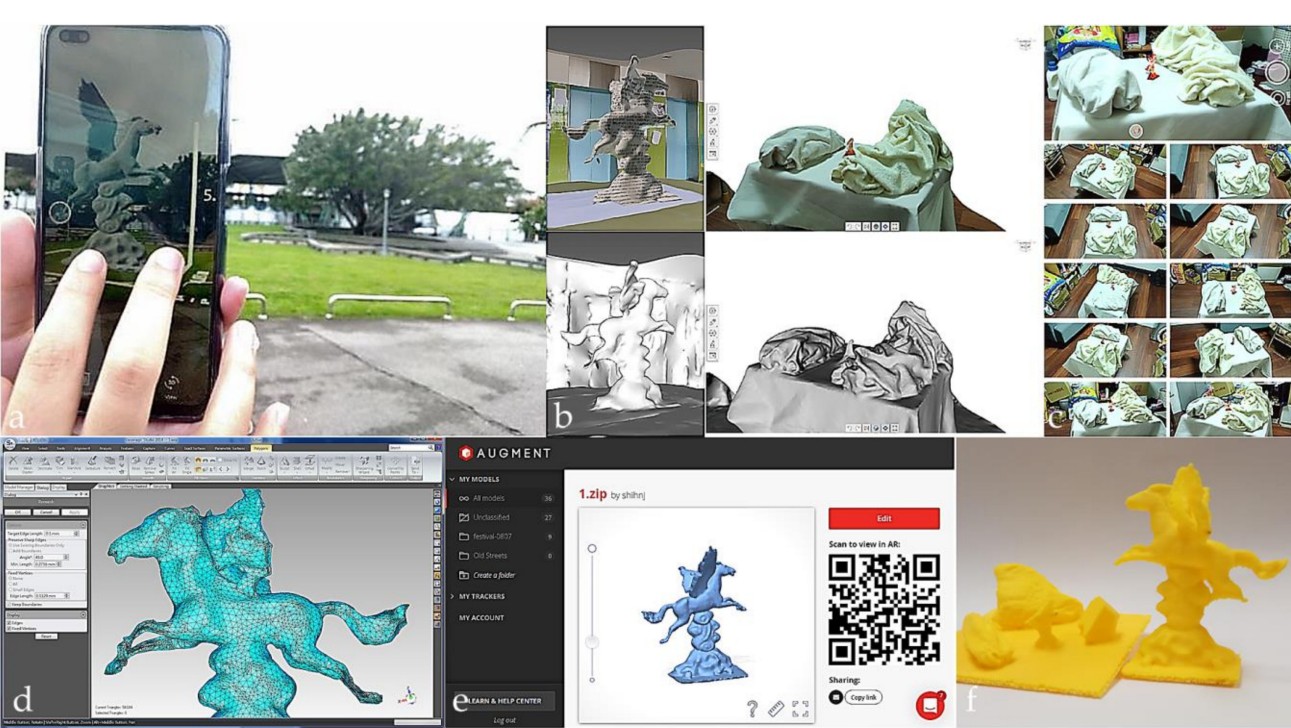

**Figure 13.** Reconstruction of the lantern in a new context as a sustainable preservation effort: (**a**) new allocated setting with the surrounding context; (**b**) photogrammetry modeling case 1; (**c**) modeling of case 2 made by the screenshots taken from the AR scenes; (**d**) the modification process; (**e**) AR QR code of the model; (**f**) 3D-printed model and layout case made by reversed reconstruction.

### 7.1. Field and Laboratory Tests

The three AR instancing scenarios mentioned above represented a simulations made from a programmer's or users' perspective. The application in Guangdong proved that the spatial structure and large staged installations can be elaborated directly from an app programmer's point of view in a park or plaza, the same as a designer could do. The application in Taipei proved that installations can be easily interpreted and improvised by researchers to assimilate into the original context like for visitors.

Although situated applications have been used to enhance or re-experience a former event, a pure application should be evolved into a reconstruction of a test object and context in architectural pedagogy or design practice. More detailed tests were conducted in the laboratory or personal working spaces where different contexts were deployed for possible design elaboration, in which a user was also a designer. The tests followed the sequence and relationship illustrated in Figure 13.

The reconstruction as a design process has to lay out the 3D model of an installation in a real environment or an artificial setting to replicate, simulate, or extend the former context. Based on the layout and complexity of the background, the context was classified into a plain background using a white cloth sheet, a more featured background using a piece of wood to simulate the landform, and a field test with a visit to a real site.

Experiments concluded the cases of success and restrictions (Table 1). The reversed reconstruction created models with different levels of detail. Most isolated subjects had models created with more detail than most of the rough models represented in a circled background, because of the depth issue or being merged into the background. Due to the interference of pedestrians during the 3D scan, some models were not created for an incomplete boundaries.

**Table 1.** Experiments in terms of field cases and design applications.

| | | Situated Tests | | | | | Results | | |
|---|---|---|---|---|---|---|---|---|---|
| | | Background | | | Arrangement | | Data | | |
| | | White Sheet | Wood | Field | Circled | None | Images | Time (min) | Model |
| field simulation: **Figure 8a, Figure 13f** | A | | | V | | V | 16 | 10 | |
| | B | | | V | | V | 33 | 10 | V |
| | C | | | V | | V | 25 | 10 | V |
| | D | | | V | | V | 46 | 10 | V |
| | E | | | V | | V | 4 | 10 | |
| | F | | | V | | V | 3 | 5 | |
| | G | | | V | | V | 7 | 5 | |
| | H | | | V | | V | 1 | 5 | |
| | I | | | V | | V | 29 | 15 | V |
| | J | | | V | | V | 2 | 5 | |
| | K | | | V | | V | 1 | 1 | |
| | L | | | V | | V | 30 | 6 | |
| | M | | | V | | V | 17 | 5 | V |
| | N | | | V | | V | 1 | 1 | |
| design simulation: **Figure 13b,c,** and below | a (lab.) | V | | | | V | 28 | 7 | V |
| | b (lab.) | | | V | | V | 74 | 10 | V |
| | c (lab.) | | | V | | V | 74 | 10 | V |
| | d (lab.) | V | | | | V | 81 | 7 | V |
| | e (lab.) | V | | | | V | 55 | 11 | V |
| | f (lab.) | V | V | | V | | 63 | 8 | |
| | g (lab.) | V | V | | V | | 32 | 5 | |
| | h (lab.) | V | V | | V | | 100 | 16 | |
| | i (lab.) | V | V | | V | | 64 | 10 | |
| | j (home) | | | V | | V | 51 | 6 | V |
| | k (home) | V | | | V | | 86 | 16 | V |
| | l (home) | V | | | V | | 64 | 12 | V |

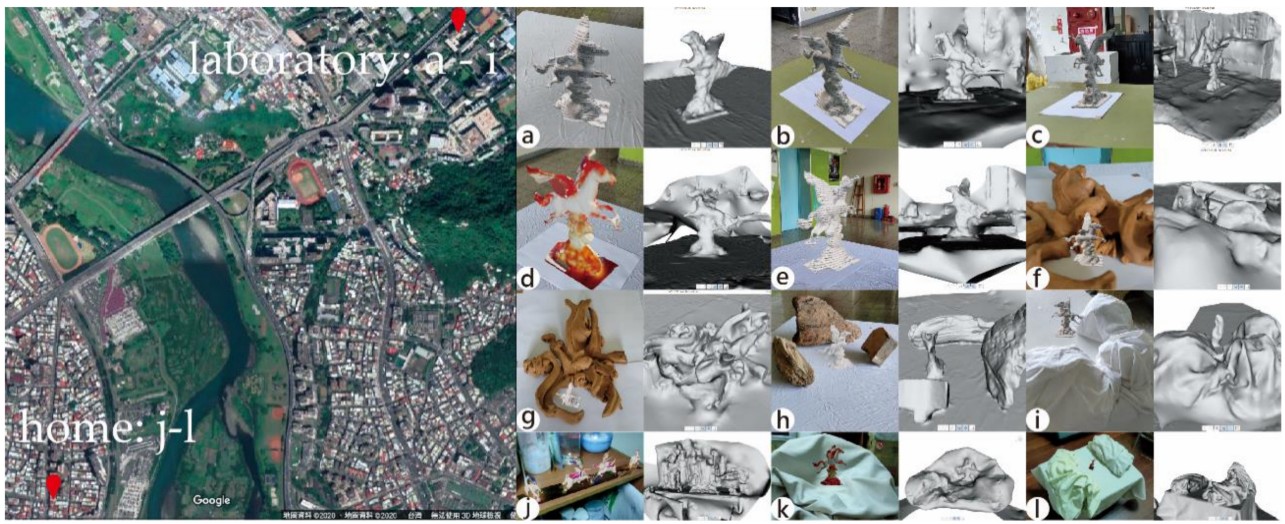

The complexity of the background contributed to the level of detail. Using the same cloth but a more complicated background, or being surrounded by woods, was more likely to create details. In a laboratory test with a white sheet acting as a simple and plain background, it was less likely that horse feet would be modeled due to the lack of reference. Models were deformed if the images were taken in an extended zoom-out mode, since conflict of depth occurred and the model always appeared in front of the wood. Success was made possible with more detail when we zoomed in on the subject by 360°. Similar results were made by circling separated pieces of wood around the AR model, to create a a vacancy for the subject.

Field tests were undertaken at the original festival site in daytime and in the evening. Researchers reported that AR models with dimensions marked on the screen were very helpful to allocate a subject with a close resemblance of size at a 1:1 scale in the field. Most of the lanterns were originally allocated in the center of a grassland. The simulation of the original setting was not successful since the soft pavement with grass waving in the wind added a drifting problem to the model. Only one model was created on the grassland. Glare from street lamps occurred and prevented taking good pictures. Therefore, a model should be allocated to avoid these problems. More interference from pedestrians occurred in the evening than in the daytime. Although adjustments had to be made for the interference, the picture-taking time remained similar.

### 7.2. Drafting-Like Assistant and Context-Based Assistant

The initial design stage without the models being placed at the correct locations can be time-consuming. The test actually took much longer to allocate all of the models in a relatively linear or perpendicular layout (Figure 14), since adjustments to specific measurements or alignments were difficult in AR. Although the model size was annotated clearly, the plan projection of the final layout exhibited a larger tolerance than anticipated. As a consequence, a good environment of interaction would require an auxiliary function for layout assistance. This concern led to the development of the app with a pre-set layout of models with which to start.

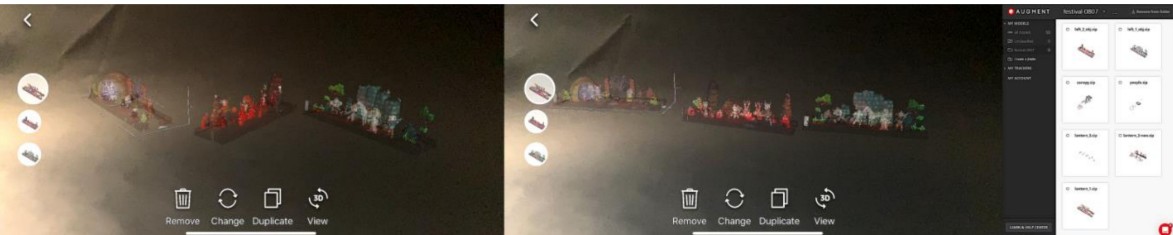

**Figure 14.** AR app for individual instancing.

### 8. Discussions

The AR application with cultural concern needs to be more deeply explored since the interaction was not limited to the manipulation of geometric instances, but also the cultural background that existed behind those instances. After all of the context-matching AR effort, it was concluded that the interaction between the AR design and the real-world case was critical, as one of them could provide a deeper comprehension of the local cultural identity. The local icon contributed a meaningful representation for the promotion process to the AR platform. The interaction experience of users under the tourism paradigm can be extended to a design practice that facilitates future urban or landscape design. The general-purposed AR instancing is also meaningful when a background is provided and related to cultural activity. The instancing can then be applied to different contexts and induce a familiar festival feeling immediately.

A traditional heritage is location-specific and cannot be relocated. The contributions can be elaborated now and symbolized in different forms of media for the creative culture industry between the traditional and modern values of time and space. The core culture activity can be promoted from the cross-city annually to cloud access all-year around. In addition to the recycling of physical lantern creations to the subsequent year, the sustainability of the lantern festival can be achieved from the scenes archived in digital format, the AR experiences of previous festivals, the construction of the spatial structure, and the 3D interaction of theme lanterns. In contrast to the richer color retrieved using photogrammetry technology, 3D scans collect more detail to differentiate parts. Both types of mesh models can be accessed from cloud data and viewed through an Internet platform. In contrast to a photogrammetry model with high reality, the level of structural detail is much greater.

The 3D sustainability of a point cloud in VR and a mesh model in AR represent an extended experience for point cloud-related application. It constituted the execution of reverse engineering applied to festival reservation with an extension to current heritage data and technology. A subjective interaction of cultural instances was made by context instancing, selective instancing, and management, such as a tool of design by AR.

### 8.1. Novelty and Contribution

The novelty and contribution of this study are three-fold: documentation, development of an AR app for situated tasks, and AR verification. The documentation was made to a special evening event, which was yet to be achieved in 3D and AR promotion. The new evolved festival of cultural heritage made this 3D record highly valuable. Moreover, both the customized and ready-made AR app were applied to situated tasks, such as promotion, comparison, and possible design evaluation in different sites. The AR not only existed as a simulation; RP-assisted AR verification was also conducted with a well-defined process to reverse-reconstruct a 3D model of the simulated scene. The feasibility of AR was extended to a new dimension, with follow-up verifications of the results.

Current representation and interaction of any artifact created in an event can be achieved through AR using customized or ready-made platforms. However, the simulation was usually accomplished in a forward manipulation, instead of backward evaluation afterward. After the tests of festival installations as fulfillment of the study goal, we would like to conduct this from a design perspective to provide a more involved and profound experience than a normal AR application can achieve.

We believe that the simulation should move to a different level of reality. With the assistance of photogrammetry modeling, images taken in the process of interaction can be referred in order to reconstruct a model. This simple approach connects a virtual world to a physical world with a feasible RP output. Although the problem of depth and deformation existed, the results here justifies the feasibility of this approach.

### 8.2. Ready-Made App and Customized Application

Trade-offs exist between the ready-made app and customized application, in which the former could create more elaborated representations of an entity and the latter had more control of the display options and a predefined relative allocation between entities. Both approaches enabled the transformation of individual entities as an extended exploration of traditional lantern craftsmanship, local folklore, and island-wide identities. However, the development effort of the latter was time- and effort-intensive.

The author developed an AR app that was feasible for simulations in a collaborative effort, since the data were created in one place and applied in another within similar or different contexts. Modification was added to enable the freedom of manipulation in rotation and scaling. For cultural promotion, ready-made apps have the mobile advantage of smartphone and cloud access to 3D models. The customized app has the mobile advantage of a predefined design layout for smartphone and cloud access. Both facilitated situated simulations.

*8.3. Revisit Experience*

The revisit experience was very interesting, and also different from the original visits that occurred years ago. An observation was made by examining if the researchers simulated the installations differently from the original settings, or how different of a change was made to the settings. The results showed both the context and the technical issues, in which the former showed that the location of installation could be easily identified by checking the resemblance of 3D scans. The latter suggested a flat and solid ground surface was more preferable and applicable in AR than a setting of grass waving in the wind. The modification of the original location can be seen from the outer boundary of the scattered pattern of test locations in Figure 8a, in which researchers did reconstruct the former experience by referring to the old axis linking the entrance and the main theme lantern.

Most of the tests were conducted by a team of two for the AR interaction and a third person for picture-taking. Researchers reported an urge to look behind the smartphone for a clearer inspection, just like an installation was actually located there. A 360 interaction to the model and scene enhanced the sense of reality. A new sense of scale was created by moving and orienting the smartphone to different parts of a huge installation, to catch the full picture of the object. The interaction made by the researchers in the air often confused pedestrians who intended to avoid interrupting, assuming pictures were being taken in front of them at that moment.

## 9. Conclusions

One of the main purposes of this article was to discuss the feasibility of the application, and the survey of user experience will be the focus of the second stage in a future study. We did not provide a specific target audience survey evaluation, except the researchers or assistants who were also the audience of past festivals over several years. The apps and 3D data were prepared based on a real event and checked if the old experience could be recalled, then moved forward to reconstruct the space. We found this novel approach valuable, it was as a looped process.

Correlation exists between an application, a simulation, a simulation that facilitates design, and a design whose outcome can be verified. This study illustrated that an additive application and a customized application could enable the festival experience in the same or at a remote site, although the procedure was not applicable to the original format of point cloud data. A novel extension was made to re-represent the results of the object and background in an AR simulation to a physical reality of the 3D model.

The three scenarios presented different levels of interaction with a scene. A forward interaction loop worked just like a normal design process, since it was in its conceptual design stage. In reality, a project may need to consider the constraints of space or installations initially and come up with a specification to be followed by a design team. The constraints included the application of constrained instance locations, re-instancing for different contexts, and a 3D scan data-originated model adaption in AR. The state of design is usually required and monitored by a project manager in a schematic design or design development stage. As a result, the management of models actually reflects how AR should work in professional practice. In fact, the way to access and interact with each model does matter. The application of LiDAR for remodeling projects is very common now. Not only does a demand exist for a platform with an app for manipulation in different hierarchies, but seamless working between the point cloud and mesh model should also be achieved.

The importance of using the festival as an instance is to prove that this is more than a desktop app. Indeed, it is a design tool and a sharable experience for others, as well as a type of as-built data that can be created and applied from different input devices, experiences, data flows, and levels of reality. The festival represents a special event that carries the development of the local culture and economy from one decade to another. The created transformation included an evolving urban boundary and tourist-oriented temporary fabric experience. In order to witness the richness of the event, the process

of reverse engineering was applied to reconstruct an as-built 3D model of the site and festival-related elements. The 3D documentation fulfills the representation needs.

**Author Contributions:** Conceptualization, methodology, validation, formal analysis, investigation, resources, 3D scan data curation, RP models, writing—original draft preparation, writing—review and editing, visualization, supervision, project administration, and funding acquisition, N.-J.S.; iOS App software, AR format transfer, validation in remote field, visualization, validation, P.-H.D.; data curation, visualization, validation, Y.-T.Q. and T.-Y.C. All authors have read and agreed to the published version of the manuscript.

**Funding:** This research was an extended study with the funding sponsored by the Ministry of Science and Technology, Taiwan, under the project number MOST 105-2221-E-011-014-MY2. The authors would like to appreciate the support.

**Conflicts of Interest:** The authors declare no conflict of interest.

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
