# Peer review of "Situated AR Simulations of a Lantern Festival Using a Smartphone and LiDAR-Based 3D Models"

_applsci, doi:10.3390/app11010012_

Round 1
Reviewer 1 Report
Authors have preseted three different scenarios with some levels of interaction with a scene.
The app is innovative, and the paper is well structured. However, from my point of view, a survey test is missing to have a feedback from different users.
I suggest the authors to perform a test and share the results inside this paper.
Author Response
Dear Reviewer:
On behalf of my co-authors, I’ll like to thank you for the significant reviewing effort. The comments are very enlightening. Hopefully I have answered all the questions. All the answers are followed by notes: (as added between line xxx – yyy) for quick search.
In the manuscript, the texts in blue color refer to the answers to all the reviewers’ comments and questions. The texts in green color refer to the authors’ editing and restructured part of the original manuscript.
Your effort is highly appreciated.
Best regards,
Naai-Jung Shih

Reviewer 2 Report
The authors of this paper have developed an AR application based on the Lantern Festival, but a detailed description of the application's design is lacking, making it difficult for the reader to determine whether the developed application has the features necessary to re-experience and redesign the festival.
In order to recreate the Intangible Cultural Heritage, the reviewer believes that there is a need for a clear description of how they solved it, organizing and describing the features that are missing in the usual AR apps.
In addition, the authors need to verify that the AR app developed by the authors achieves the desired purpose. The current paper includes a description of limited functionality with pictures of the screen. However, the reviewers believe that a more detailed description of the features and an explanation of the rationale for its design is needed. 
The reviewer thinks that both subjective and objective evaluations by the target audience of this application are also necessary to justify that the design of this application works well.
In addition to the above comments, the reviewers found the paper to be redundant in parts, so the reviewer recommends that the authors polish the entire paper and remove as much of the duplicate content as possible.
Proper nouns such as Sketchfab, Augment, Depth API, Cloud Compare, Meshlab, etc., referring to various AR platforms and tools, are used in this paper, and the reviewers think that these need to be cited as well.
When referring to figures (left) and (right) are ambiguous. Please label them (a), (b), etc. to make them clearer.
The text in the figures in Figures 1, 3, 7, 8, 9, and 10 is too small to read.
Figure 4 is described in l.208-212, but the figure caption is not consistent with the text in the paper. The same is true for Figure 5.
Author Response

(The authors gave the same response as above.)

Reviewer 3 Report
I found in your article an impressive work, which required multidisciplinary knowledge combined in a unitary approach.
The research was very ambitious and is clearly described, although sometimes the reader may be lost due to the multitude of technical details.
The article can be appreciated for the remarkable effort of scanning the scene, recording point clouds, decimating and all the other steps.
As observations, in the abstract I found a phrase that does not seem very clear to me:
“Three AR instancing scenarios were applied to the 12 converted scanned data from an interaction to the entire site, a forward additive instancing and 13 interaction, to interactions with a pre-defined model layout.”
Also, the figures are unclear, and the text cannot be understood. I would suggest restructuring the way the screenshots are presented so that the elements you are referring to be more visible.
I did not find anything about how the application was perceived by users in order to have a measure of the intention to use such an application. Did you not perform tests with users? It would be good to add some opinions, even if the purpose of the article is to present only the application itself.
Author Response

(The authors gave the same response as above.)
